# Conventional video-based system for measuring the subtask speed of the Timed Up and Go Test in older adults: Validity and reliability study

**Teerawat Kamnardsiri**[1,2], **Nuanlaor Thawinchai**[3], **Arisa Parameyong**[3], **Pim Pholjaroen**[3], **Khanittha Wonglangka**[3], **Paphawee Prupetkaew**[3], **Sirinun Boripuntakul**[2,3]*

1 Department of Digital Game, College of Arts, Media and Technology, Chiang Mai University, Chiang Mai, Thailand, 2 Research Group in Informatics for Well-being Society, Chiang Mai University, Chiang Mai, Thailand, 3 Faculty of Associated Medical Sciences, Department of Physical Therapy, Chiang Mai University, Chiang Mai, Thailand

* sirinun.b@cmu.ac.th

**Data Availability Statement:** All relevant data are within the paper and its Supporting Information files.

## Abstract

The Timed Up and Go Test (TUG) is a simple fall risk screening test that covers basic functional movement; thus, quantifying the subtask movement ability may provide a clinical utility. The video-based system allows individual's movement characteristics assessment. This study aimed to investigate the concurrent validity and test-retest reliability of the video-based system for assessing the movement speed of TUG subtasks among older adults. Twenty older adults participated in the validity study, whilst ten older adults participated in the reliability study. Participant's movement speed in each subtask of the TUG under comfortable and fast speed conditions over two sessions was measured. Pearson correlation coefficient was used to identify the validity of the video-based system compared to the motion analysis system. Intraclass correlation coefficient (ICC3,2) was used to determine the reliability of the video-based system. The Bland-Altman plots were used to quantify the agreement between the two measurement systems and two repeatable sessions. The validity analysis demonstrated a moderate to very high relationship in all TUG subtask movement speeds between the two systems under the comfortable speed ($r = 0.672$–$0.906$, $p < 0.05$) and a moderate to high relationship under the fast speed ($r = 0.681$–$0.876$, $p < 0.05$). The reliability of the video-based system was good to excellent for all subtask movement speeds in both the comfortable speed (ICCs = $0.851$–$0.967$, $p < 0.05$) and fast speed (ICCs = $0.720$–$0.979$, $p < 0.05$). The Bland-Altman analyses showed that almost all mean differences of the subtask speed of the TUG were close to zero, within 95% limits of agreement, and symmetrical distribution of scatter plots. The video-based system was a valid and reliable tool that may be useful in measuring the subtask movement speed of TUG among healthy older adults.

**Funding:** -Teerawat Kamnardsiri -College of Arts, Media and Technology, Chiang Mai University, Thailand and Research Group of Modern Management and Information Technology, College of Arts, Media and Technology, Chiang Mai University, Thailand. -The funders had no role in study design, data collection and analysis, decision to publish, or preparation of the manuscript.

**Competing interests:** The authors have declared that no competing interests exist.

## Introduction

Aging is a complex phenomenon associated with a progressive physical decline in multiple systems, such as the sensory, neuromuscular, and cognitive systems, predisposing mobility problems and falls [1]. Given that older adults are susceptible to falling, functional screening is a critical process for allowing fall prevention among this population. Among performance-based tests for fall risk screening, the Timed Up and Go Test (TUG) is a simple test commonly used to determine mobility with correlates to balance and fall risk across various older adult populations and settings [2,3]. This test requires a person to stand from a seated position, walk three meters forward, turn, walk back to the chair, and sit down again, with the outcome being time to completion using a stopwatch. A faster time designates a better functional performance in transferring, walking, and turning movements [4]. The existing studies suggest that the time taken to complete the test had good reliability and validity in both community-dwelling older adults and specific conditions such as stroke, Parkinson's disease, and Alzheimer's disease [5–8].

Currently, studies suggest that a major constraint in using the total TUG time is that it only provides a global mobility performance, failing to provide data exploring the movement control subtasks (i.e., sit-to-stand, walking, turning, and stand-to-sit) [9,10]. Understanding the TUG's respective subcomponent speeds would provide additional meaningful information regarding clinical interpretation and specific therapeutic guidance for healthcare providers. For example, among older adults with poor TUG performance, it is noteworthy to quantify whether a specific subtask's performance is due to slow chair activity, slow walking, slow turning, or if the time is simply due to general slowness for all components. In practice, however, measuring each subtask speed of TUG has been difficult to administer by a conventional hand-time stopwatch. Several recent studies have explored TUG subcomponents using sensor technology, such as wearable inertial sensors [9,11], smartphones [11,12], and ambient sensors [11,13]. However, there are some challenges such as high costs, susceptibility to interference and damage, and limited range of use [9,11–14]. The video-based system has received increasing attention in the movement analysis field. The video-based system, a markerless approach, is one of the more flexible ways of data acquisition that allows participants to move naturally under various environmental conditions. Extant studies examining the TUG subtasks using the conventional video-based system have been scarce. Previous studies have developed an algorithm method for the automated segmentation of the TUG subcomponents using a simple video-based system [15,16]. However, based on our knowledge, there still needs to be more evidence related to the validity and reliability of the conventional video-based system for tracking TUG subtask performances among older adults. Therefore, this study aimed to determine the concurrent validity and test-retest reliability of the conventional video-based system for measuring the movement speeds of the TUG subcomponents in community-dwelling older adults. Specifically, the concurrent validity of the movement speed of each TUG subtask is also examined concerning a gold standard as the motion analysis system.

## Materials and methods

### Participants

Twenty community-dwelling older adults living in the Chiang Mai province, Thailand, participated in the study. Eligible participants were included if they were 60 years or older, able to walk safely without assistive devices or another person's assistance, and able to comprehend and follow the instructions. Participants were excluded if they had any of the following conditions that would affect movement performance: (i) major cognitive impairment (defined as a

score less than 23 points or depending on the level of education on the Mental State Examination T10; MSET10) [17], (ii) acute or chronic neurological conditions (e.g., Parkinson's disease, stroke), (iii) severe musculoskeletal impairments (e.g., joint pain, deformity), (iv) unstable health conditions (e.g., hypertension, asthma, heart disease), (v) depressive symptoms (determined as a score of more than 6 points on the Thai Geriatric Depression Scale-15; TGDS-15) [18], (vi) uncorrected visual or hearing impairments, and (vii) current use of sedative or antipsychotic drugs. The study protocol was approved by the Human Ethical Review Board of the primary investigator's institution (approval number: AMSEC-63EX-073). All participants gave written informed consent before enrollment.

## Set up of the video-based system and motion analysis system

The three-dimensional motion analysis system (Motion Analysis®, Motion Analysis Corporation, Santa Rosa, California, USA) with ten infrared flashlight cameras (Eagle-4 camera system, Motion Analysis®) and EVaRT 5.0 software was used to capture the subtask speeds of TUG. The sampling rate of the motion analysis system was set at 120 Hz. Moreover, the x, y, and z marker coordinates data were processed and filtered with a fourth-order, low-pass Butterworth filter with a 6 Hz cut-off frequency. Each TUG subtask speed was analyzed using custom-written programs in MATLAB 2015a (The MathWorks, Inc., Natick, Massachusetts, USA). In the present study, the capture volume was 6 meters long and 5 meters wide, which allowed 30 square meters for data collection.

Concerning the video-based system setup, a single high-definition (HD) video camera recorder (SONY Mirrorless SLR α5100, Sony, Japan) was located to the side and in the middle of the TUG walkway and perpendicular to the plane of the participant's movement. The single digital video camera was set at a height of 1.0 meters above the ground and a distance of 5.0 meters from the testing walkway. The video rate was set at 60 Hz with a resolution of $1280 \times 720$ pixels (MOV format). The markers were positioned 3 meters apart at the starting point ($M_2$) and the end of the 3-meter walkway ($M_1$). In addition, another two markers were set 0.5 meters apart on the floor beside the chair leg ($M_3$) and the chair armrest ($M_4$). The external synchronization box was located beside the chair. The perspective view of the TUG test setup is illustrated in Fig 1.

## Assessment of the concurrent validity and test-retest reliability of the video-based system

For descriptive purposes, each participant was interviewed about demographic data such as age, weight, height, and fall history in the previous 12 months. Prior to performing the TUG, participants were asked to wear a motion capture suit and their regular footwear. The reflective markers (2.0 cm diameter) were bilaterally placed over the participant's anterior superior iliac spine (ASIS) and posterior superior iliac spine (PSIS). Following the TUG protocol, participants were seated on a standard armchair placing their back against the chair and resting their arms on the armrest. Participants were subsequently instructed to rise from the chair in response to the command "Go", walk 3 meters to a mark placed on the floor, turn around at the 3-meter mark, walk back to the chair, and sit down again [19]. Participants were allowed to complete one practice trial to familiarize themselves with the test.

Twenty participants performed the TUG with two different walking speed conditions, including walking at a comfortable speed for two trials and walking at a fast speed for two trials (a total of four walking trials). These trials were performed sequentially with a 1-minute resting period between each trial. The motion analysis and video-based systems were used to simultaneously capture participants' movements while performing the test throughout each trial. The

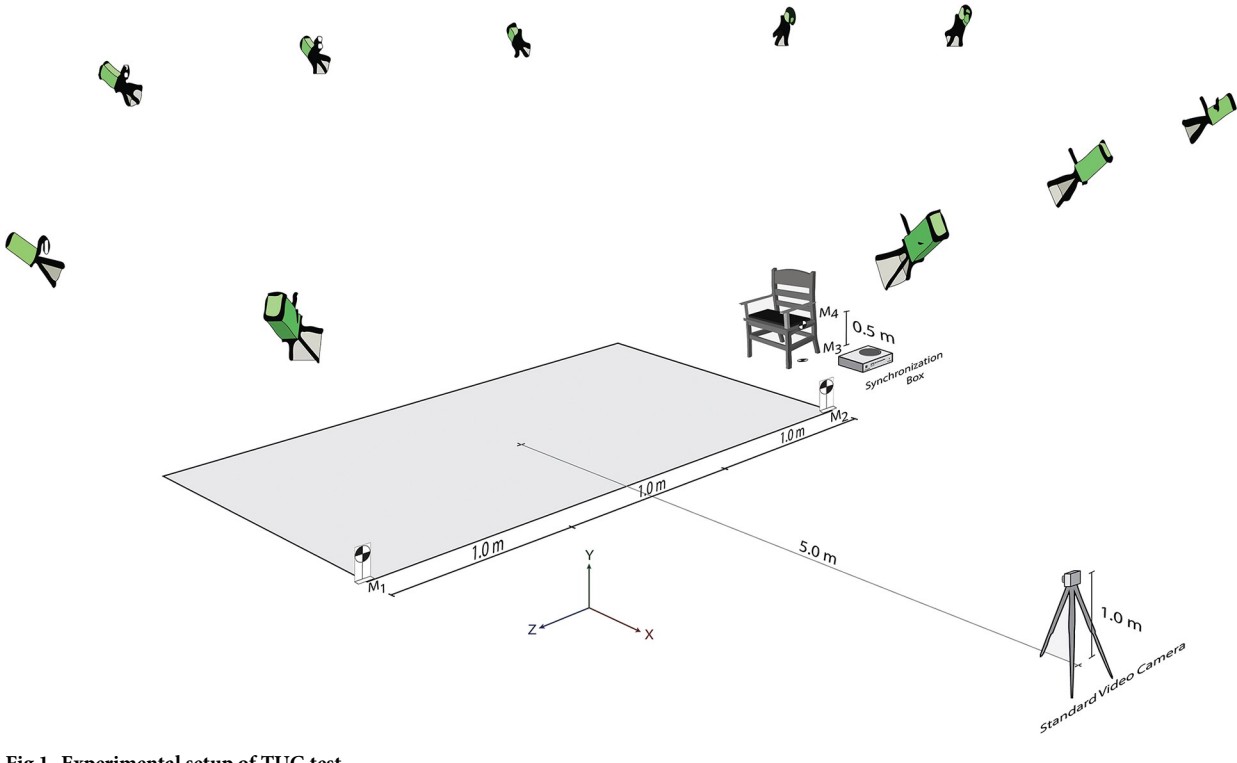

**Fig 1. Experimental setup of TUG test.**

external synchronization box was employed to identify the starting and stopping events during the video recording in accordance with the motion capture to determine the study's concurrent validity.

Among twenty participants, ten were selected based on availability to investigate the test-retest reliability. Participants performed the TUG at a comfortable speed and a fast speed in two sessions on the same day with a 30-minute interval between sessions. During the 30-minute period, participants were asked to rest to avoid the fatigue effect. In the present study, three well-trained assessors, including one motion capture specialist and two physical therapists, administered the motion capture, the video capture, and the TUG test throughout data collection.

## Video-based system for processing the TUG subtask speeds

The processing method of our developed video-based system for quantifying the TUG subtask speeds was categorized into six steps as follows (Fig 2):

**The first step** was calibrating the overall capture volume process using the markers labeled $M_1$ to $M_4$ (Fig 1). $M_1$ and $M_2$, located 3 meters apart at the beginning and the end of the testing walkway, were used to calculate the capture volume in the horizontal plane ($x$-axis). Moreover, $M_3$ and $M_4$, located 0.5 meters apart on the floor and the armrest, were used to calculate the capture volume in the vertical plane ($y$-axis).

**The second step** was human body region detection. The background subtraction technique was employed to detect the moving objects (i.e., participants) within the video frames as the foreground image *(Fg)* compared to a reference static screen as the background image *(Bg)*. The background subtraction is given by Eq (1).

$$BgSubtraction(x, y) = Fg(x, y) - Bg(x, y) \qquad (1)$$

In this phase, the moving RGB image was converted into a grayscale image and then into binary imagery using the Thresholding Operation denoted in Eq (2). The reference static frame was also transformed into a binary image to produce a clear background. Afterwards, the foreground video was subtracted from the background screen in each frame.

$$HumanBody(x, y) = \begin{cases} 1 & if\ BgSubtraction(x, y) \leq \tau \\ 0 & if\ BgSubtraction(x, y) > \tau \end{cases} \quad (2)$$

Where $\tau$ = Threshold value = 50.

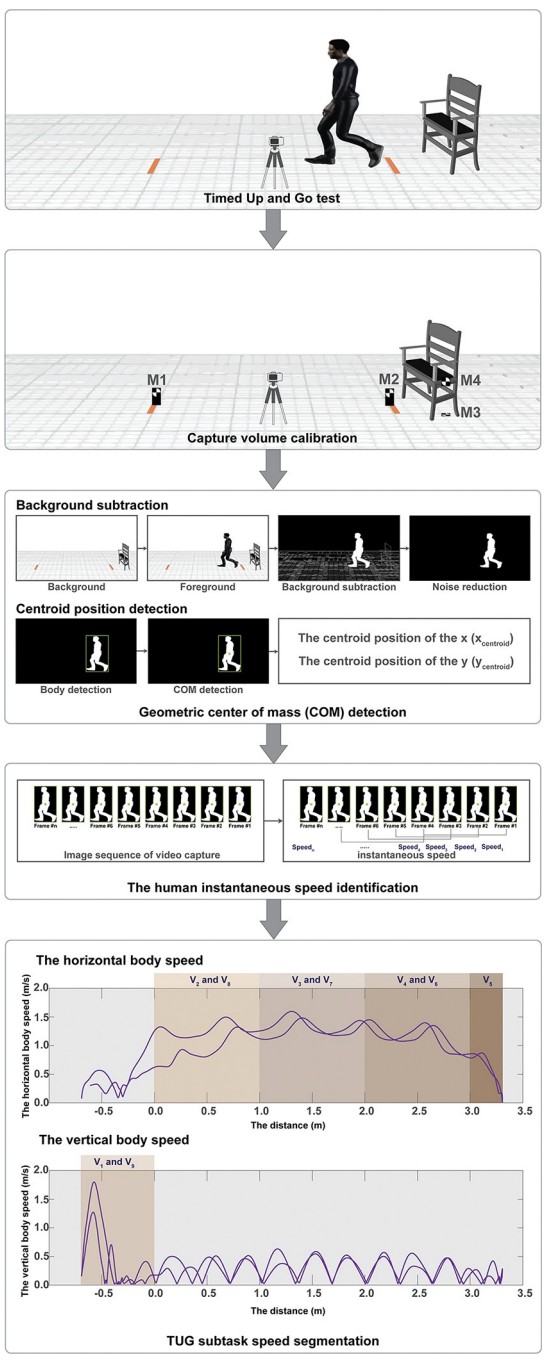

**Fig 2. The diagram of the video-based system processing for the TUG subtask speed detection.**

To reduce uncertain objects (i.e., image noises), a two-dimensional median filter ($5 \times 5$ adjacent pixels) was used to remove noise from the *BgSubtraction* image and then converted to a binary image (black: '0' and white: '1').

**The third step** was detecting the geometric center of mass (COM) position (labeled as 'centroid') of the human body. The centroid of each participant was tagged to calculate the position in both horizontal and vertical directions using a two-dimensional lamina [20].

*BodyPosition* ($x_{centroid}, y_{centroid}$), $x_{centroid}$ and $y_{centroid}$ of $n$ point masses ($m_i$) located at positions $x$: ($x_i$) and position $y$: ($y_i$) were given by Eqs (3–5).

$$x_{centroid} = \frac{\sum_{i=1}^{n} m_i x_i}{M} \tag{3}$$

and

$$y_{centroid} = \frac{\sum_{i=1}^{n} m_i y_i}{M} \tag{4}$$

where,

$$M = \sum_{i=1}^{n} m_i = total\ mass \tag{5}$$

**The fourth step** was the identification of the human body position. To trace the centroid of the human body while performing the test, the calibration values (obtained from $M_1$, $M_2$, $M_3$, and $M_4$) were used to define the centroid position of the $x$ ($x_{centroid}$) and $y$ ($y_{centroid}$) axes in the image pixels. Subsequently, the position of the axes ($x,y$) was transformed to the actual centroid position in the world coordinate system in the horizontal direction (*Actual_X$_{Position}$*) as given by Eqs (6–9).

$$CalVolume\_X = M_2 - M_1 \tag{6}$$

$$OnePixel\_X_{Distance} = \frac{3}{CalVolume\_X} \tag{7}$$

where, 3 is the actual distance between $M_1$ and $M_2$ (3.0 m)

$$Body\_X_{pixel} = x_{centroid} - M_1 \tag{8}$$

$$ActualPos\_Y = Body\_Y_{pixel} \times OnePixel\_Y_{Distance} \tag{9}$$

And also, the vertical direction (*Actual_Y$_{Position}$*) as given by Eqs (10–13).

$$CalVolume\_Y = M_4 - M_3 \tag{10}$$

$$OnePixel\_Y_{Distance} = \frac{0.5}{CalVolume\_Y} \tag{11}$$

where, 0.5 is the actual distance between $M_3$ and $M_4$ (0.5 m).

$$Body\_Y_{pixel} = M_3 - y_{centroid} \tag{12}$$

$$ActualPos\_Y = Body\_Y_{pixel} \times OnePixel\_Y_{Distance} \tag{13}$$

**The fifth step** was the identification of the human instantaneous speed. The centroid

position in the coordinate system was collected in array form. The instantaneous movement speed in the horizontal plane ($Speed_{inx}$) was calculated in every four frames ($DurationTime = 4$) of the video image as given by Eqs (14–16).

$$speed_{inx} = \frac{\Delta x_{inx}}{\Delta tx_{inx}} \qquad (14)$$

where,

$$\Delta x = distance_{(inx+DurationTime)} - distance_{inx} \qquad (15)$$

and

$$\Delta tx = time_{(inx+DurationTime)} - time_{inx} \qquad (16)$$

Additionally, the instantaneous movement speed in the vertical plane ($Speed_{iny}$) was given by Eqs (17–19).

$$Speed_{iny} = \frac{\Delta y_{iny}}{\Delta ty_{iny}} \qquad (17)$$

where,

$$\Delta y = distance_{(iny+DurationTime)} - distance_{iny} \qquad (18)$$

and

$$\Delta ty = time_{(iny+DurationTime)} - time_{iny} \qquad (19)$$

**The sixth step** was the TUG subtask speed segmentation. The movement speed when performing the TUG was split into nine subtasks (Fig 3), including

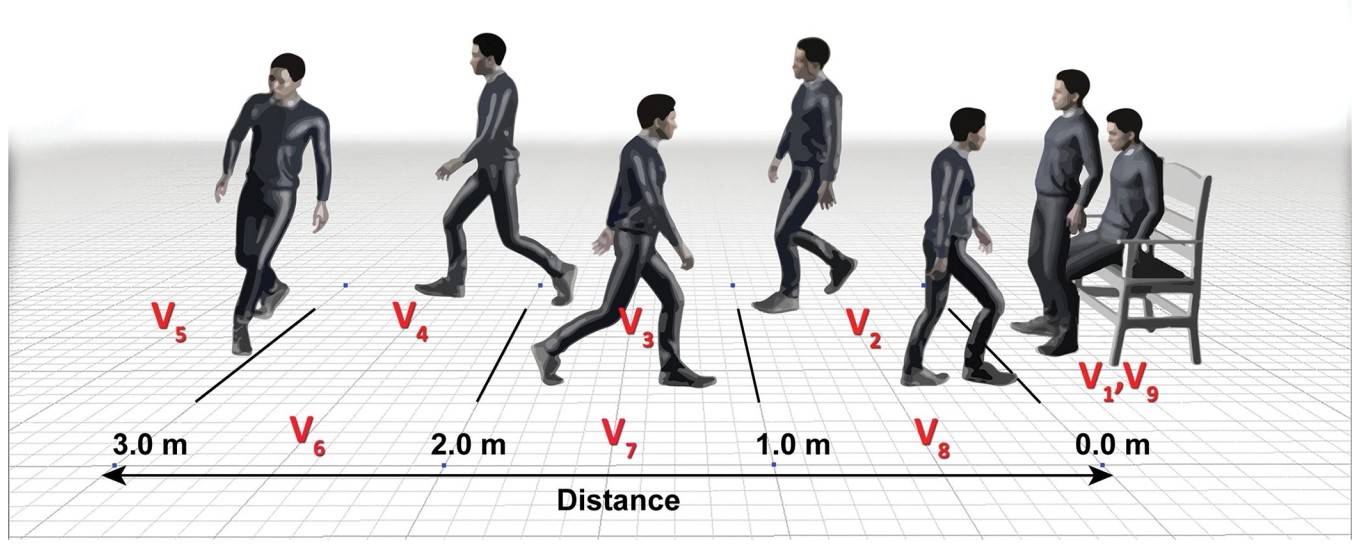

**Fig 3. The speed of TUG subtasks including $V_1$: Sit-to-stand speed, $V_{2\ to}\ V_4$: Walk forward speed, $V_5$: Turn around speed, $V_{6\ to}\ V_8$: Walk back speed, and $V_9$: Stand-to-sit speed.**

i. **Sit-to-stand ($V_1$)** was calculated using the movement distance in the vertical plane of the centroid point and the number of video frames between the sitting and standing positions.

ii. **Walk forward meter 0 to 1 ($V_2$)** was calculated from the movement distance in the horizontal plane of the centroid point and the number of video frames from meter 0 to meter 1.

iii. **Walk forward meter 1 to 2 ($V_3$)** was calculated from the movement distance in the horizontal plane of the centroid point and the number of video frames from meter 1 to meter 2.

iv. **Walk forward meter 2 to 3 ($V_4$)** was calculated from the movement distance in the horizontal plane of the centroid point and the number of video frames from meter 2 to meter 3.

v. **Turn around ($V_5$)** was calculated from the movement distance in the horizontal plane of the centroid point and the number of video frames from 3-meter walking forward and 3-meter walking back.

vi. **Walk back meter 3 to 2 ($V_6$)** was calculated from the movement distance in the horizontal plane of the centroid point and the number of video frames from meter 3 to meter 2.

vii. **Walk back meter 2 to 1 ($V_7$)** was calculated from the movement distance in the horizontal plane of the centroid point and the number of video frames from meter 2 to meter 1.

viii. **Walk back meter 1 to 0 ($V_8$)** was calculated from the movement distance in the horizontal plane of the centroid point and the number of video frames from meter 1 to meter 0.

ix. **Stand-to-sit ($V_9$)** was calculated from the movement distance in the vertical plane of the centroid point and the number of video frames between the standing and sitting positions.

The geometric centroid position of each participant and the TUG subtask segmentations were quantified using a MATLAB 2015a (The MathWorks, Inc., Natick, Massachusetts, USA) script with the computer vision and image processing toolbox. The Windows 10 OS laptop computer equipped with Intel (R) Core (TM) i5-8265U CPU@1.60 GHz, 2 GB NVIDIA graphic card, and 8 GB DDR4 RAM (ASUSTek Computer Inc., Taipei, Taiwan) was employed to calculate the data.

## Motion analysis system for processing the TUG subtask speeds

The COM of each participant ($COM_{MoCap}$) was obtained from an intersection line of four markers (i.e., right ASIS: R_ASIS, left ASIS: L_ASIS, right PSIS: R_PSIS, and left PSIS: L_PSIS). $COM_{MoCap}$ was given by Eq (20) [20].

$$COM_{MoCap}(x', y', z') = \left( \frac{\sum_{i=1}^{n} x_i}{n}, \frac{\sum_{i=1}^{n} y_i}{n}, \frac{\sum_{i=1}^{n} z_i}{n} \right) \tag{20}$$

where,

$$x = \{R\_AISI_x, L\_PSIS_x, R\_PSIS_x, L\_ASIS_x\},$$
$$y = \{R\_AISI_y, L\_PSIS_y, R\_PSIS_y, L\_ASIS_y\},$$
$$z = \{R\_PSIS_z, L\_PSIS_z, R\_PSIS_z, L\_ASIS_z\}$$

and $n$ is the number of reflective markers (i.e., four markers). Regarding the movement of $COM_{MoCap}$, TUG was segmented into nine subtasks:

i. **Sit-to-stand** was the $COM_{MoCap}$ speed in the vertical plane between the sitting and standing positions.

ii. **Walk forward** was the $COM_{MoCap}$ speed in the horizontal plane at each meter (i.e., meters 0 to 1, 1 to 2, 2 to 3).

iii. **Turn around** was the $COM_{MoCap}$ speed in the horizontal plane from 3-meter walking forward and 3-meter walking back.

iv. **Walk back** was the $COM_{MoCap}$ speed in the horizontal plane at each meter (i.e., meters 3 to 2, 2 to 1, 1 to 0).

v. **Stand-to-sit** was the $COM_{MoCap}$ speed in the vertical plane between the standing and sitting positions.

## Synchronization method between the video-based system and motion analysis system

The synchronization box (patent application number: 2203003001) mechanic consisted of lighting, a reflective marker, and a pressure plate switch (Fig 4). The red light was illuminated when the participant sat on the chair and was turned off when one stood up (the starting point of a video-based system). When the participant returned to sit down again, the red light was turned on (the stopping point of the video-based system). Moreover, the reflective marker was placed on a movable disc that was controlled by a motor. The disc moved when the participant sat on the chair and was stopped when one raised up (the starting point of the motion analysis system). When the participants sat down on the chair again, the disc was stopped (the stopping point of the motion analysis system). In this way, the synchronization box was used to synchronize the starting and stopping points of both systems.

## Statistical analysis

The concurrent validity of the video-based system compared to the motion analysis system in detecting the TUG subtask speeds was established using the Pearson correlation coefficient. A probability level of 0.05 was set to denote significance. The relative agreement between the two systems was interpreted as follows: r < 0.50 was low, r = 0.50 to 0.69 was moderate, r = 0.70 to 0.89 was high, and r > 0.90 was very high [21]. The test-retest reliability was determined for each TUG subtask using a 2-way mixed effects intraclass correlation coefficient model (ICC3,2) and corresponding 95% confidence intervals (CI). The ICC was defined as follows:

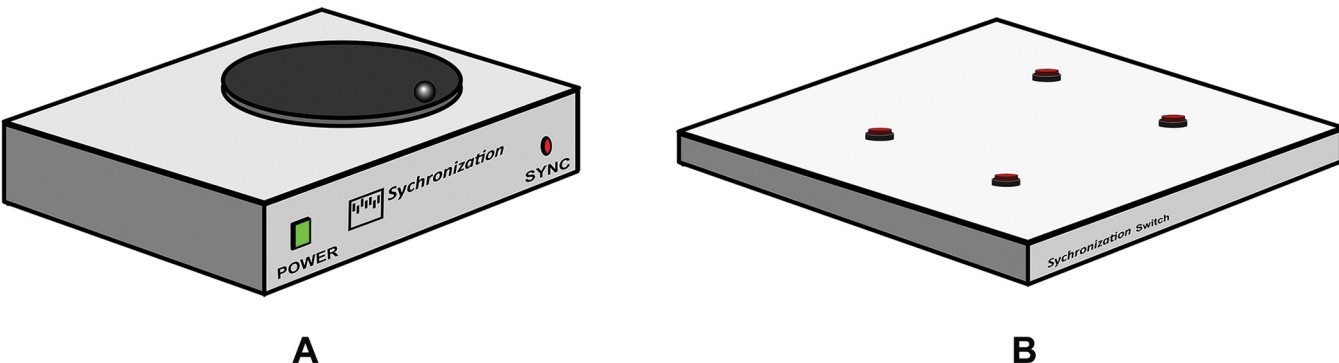

**Fig 4.** Synchronization device: (A) external synchronization box and (B) switch for triggering the starting point and ending point of the TUG.

ICC < 0.50 was poor, ICC = 0.50 to 0.69 was moderate, ICC = 0.70 to 0.89 was good, and ICC ≥ 0.9 was excellent [22]. The Bland-Altman plots were employed to illustrate the agreement between the two systems (i.e., video-based system and motion analysis system) and the first and second sessions (i.e., session 1 and session 2) [23]. The 95% limits of agreement (LOA), mean difference ± 1.96 standard deviation (SD), were used to determine the natural variation of the data in which the narrow LOA designates high stability [24]. The IBM SPSS software (version 21.0, IBM Corporation, Armonk, NY) was used to perform the statistical analyses.

## Results

### Participant characteristics

The demographic characteristics of the participants are summarized in Table 1. All participants were recruited from the community setting. Twenty participants (5 male and 15 female) with a mean age of 66.90 (5.51) years participated in the validity study. Twelve older adults reported no history of falls, while three participants reported falling once during the past year. Among twenty participants, ten participants (2 male and 8 female) with a mean age of 63.60 (2.95) years underwent the reliability study. Nine participants did not have a history of falls, whereas one had a single fall in the past year.

### Concurrent validity

The correlation between the video-based system and motion analysis system in the nine TUG subtasks in the comfortable speed condition ranged from 0.672 to 0.906, indicating a moderate to very high correlation. Specifically, the lowest correlation was found in the sit-to-stand subtask (r = 0.672, p < 0.001) and the highest correlation was observed in the walk forward from meter 1 to 2 subtask (r = 0.906, p < 0.0001). As for the fast speed condition, the correlation between the two systems for the nine TUG subtasks ranged from 0.681 to 0.876, indicating a moderate to high relationship. Like the comfortable speed condition, the lowest correlation was found in the sit-to-stand subtask (r = 0.681, p < 0.001), whereas the highest agreement

**Table 1. Demographic characteristics of the participants.**

| Characteristic | Means (SD) | Range (min-max) |
|---|---|---|
| **Validity study (N = 20)** | | |
| Age (years) | 66.90 (5.51) | 60.00–80.00 |
| Height (cm) | 154.95 (8.64) | 138.00–168.00 |
| Weight (kg) | 56.80 (12.02) | 39.00–89.00 |
| Body mass index (kg.m$^{-2}$) | 23.54 (3.96) | 17.80–36.11 |
| MSET10 (score, total score = 0–29) | 26.25 (1.74) | 23.00–29.00 |
| TGDS-15 (score, total score = 0–15) | (2.02) | 0.00–8.00 |
| **Reliability study (N = 10)** | | |
| Age (years) | 63.60 (2.95) | 60.00–70.00 |
| Height (cm) | 156.60 (7.26) | 147.00–168.00 |
| Weight (kg) | 61.20 (14.05) | 39.00–89.00 |
| Body mass index (kg.m$^{-2}$) | 24.80 (4.82) | 17.80–36.11 |
| MSET10 (score, total score = 0–29) | 26.10 (1.79) | 23.00–29.00 |
| TGDS-15 (score, total score = 0–15) | (2.67) | 0.00–8.00 |

MSET10 = Mental State Examination T10; TGDS-15 = Thai Geriatric Depression Scale-15.

**Table 2. Correlation for the TUG subtask measures, as determined by the video-based system against those obtained from the motion analysis system.**

| TUG subtasks | Movement speed (m/s) | | r (95% CI) | p-value |
|---|---|---|---|---|
| | Video-based system Mean (SD) | Motion analysis system Mean (SD) | | |
| **Comfortable speed** | | | | |
| $V_1$ (sit-to-stand) | 0.342 (0.075) | 0.354 (0.069) | 0.672 | 0.001[a] |
| $V_2$ (walk meter 0–1) | 0.951 (0.121) | 1.127 (0.138) | 0.851 | 0.0001[b] |
| $V_3$ (walk meter 1–2) | 1.075 (0.159) | 1.208 (0.141) | 0.906 | 0.0001[b] |
| $V_4$ (walk meter 2–3) | 0.883 (0.171) | 0.967 (0.104) | 0.834 | 0.0001[b] |
| $V_5$ (turn around) | 0.385 (0.096) | 0.430 (0.107) | 0.791 | 0.0001[b] |
| $V_6$ (walk meter 3–2) | 0.974 (0.131) | 1.078 (0.120) | 0.747 | 0.0001[b] |
| $V_7$ (walk meter 2–1) | 1.050 (0.126) | 1.171 (0.125) | 0.756 | 0.0001[b] |
| $V_8$ (walk meter 1–0) | 0.704 (0.081) | 0.877 (0.123) | 0.764 | 0.0001[b] |
| $V_9$ (stand-to-sit) | 0.216 (0.061) | 0.264 (0.072) | 0.773 | 0.0001[b] |
| **Fast speed** | | | | |
| $V_1$ (sit-to-stand) | 0.389 (0.077) | 0.405 (0.077) | 0.681 | 0.001[a] |
| $V_2$ (walk meter 0–1) | 1.130 (0.138) | 1.369 (0.172) | 0.876 | 0.0001[b] |
| $V_3$ (walk meter 1–2) | 1.303 (0.192) | 1.467 (0.173) | 0.872 | 0.0001[b] |
| $V_4$ (walk meter 2–3) | 1.019 (0.203) | 1.128 (0.175) | 0.756 | 0.0001[b] |
| $V_5$ (turn around) | 0.416 (0.124) | 0.451 (0.132) | 0.862 | 0.0001[b] |
| $V_6$ (walk meter 3–2) | 1.161 (0.186) | 1.265 (0.176) | 0.810 | 0.0001[b] |
| $V_7$ (walk meter 2–1) | 1.280 (0.177) | 1.438 (0.173) | 0.829 | 0.0001[b] |
| $V_8$ (walk meter 1–0) | 0.876 (0.128) | 1.106 (0.189) | 0.733 | 0.0001[b] |
| $V_9$ (stand-to-sit) | 0.264 (0.084) | 0.313 (0.083) | 0.714 | 0.0001[b] |

r = Correlation coefficients, CI = Confidence interval.

[a]Significant difference at $p < 0.001$

[b]Significant difference at $p < 0.0001$.

was observed in the walking forward from meter 0 to 1 subtask ($r = 0.876$, $p < 0.0001$). The correlation between the video-based and the motion analysis systems and the mean values of movement speed in each TUG subtask are illustrated in Table 2. The Bland-Altman plots for the subtask speeds of the TUG at the comfortable and fast speed conditions in both systems are displayed in Figs 5 and 6. For almost every participant across the subtasks of TUG, the mean differences between the two systems were close to zero within 95% LOA and symmetrical scatter plot distribution, suggesting good agreement. There was one outlier (5%) across seven subtasks of TUG in the comfortable speed condition (Fig 5) and one outlier (5%) for two subtasks, and two outliers (10%) for three subtasks of TUG in the fast speed condition (Fig 6).

## Test-retest reliability

As for the comfortable speed condition, the test-retest reliability of the video-based system for measuring the nine TUG subtask movement speeds was good to excellent (ICCs = 0.851 to 0.967, $p < 0.05$). In addition, ICCs for all TUG subtasks in the fast speed condition between sessions were good to excellent (ICCs = 0.720 to 0.979, $p < 0.05$). The test-retest reliability of the video-based system for determining subtask movement speed is presented in Table 3. Figs 7 and 8 illustrate the Bland-Altman plots for the subtask movement speeds of TUG under both the comfortable and fast speed conditions. The mean differences between the first and second sessions in all TUG subtask movement speed for almost all participants approached zero. In

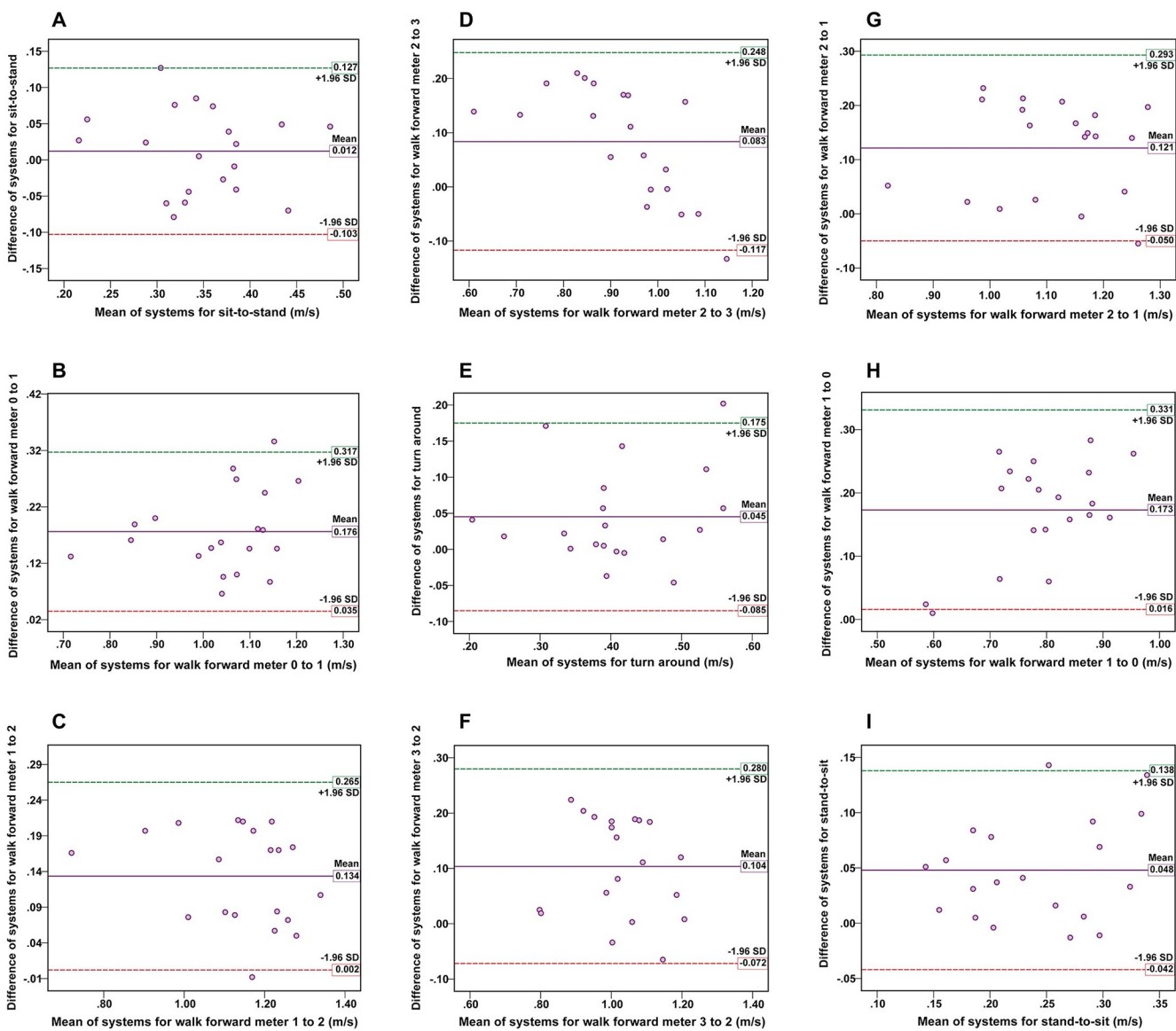

**Fig 5. The Bland-Altman plots in the concurrent validity assessment for the comfortable speed condition.** The plots display the agreement between the two systems for measurements of the subtask speed of the TUG: (A) sit-to-stand, (B) walk meter 0–1, (C) walk meter 1–2, (D) walk meter 2–3, (E) turn around, (F) walk meter 3–2, (G) walk meter 2–1, (H) walk meter 1–0, and (I) stand-to-sit. The x-axis represents the mean values, and the y-axis represents the mean difference between the two systems for each subtask of TUG. Reference lines show the mean difference between the two systems (solid line) and 95% LOA for the mean difference (dash line).

addition, the symmetrical distribution of the scatter plot graph and narrow LOA showed strong reliability levels. Specifically, one outlier (10%) was found in eight subtasks of TUG under the comfortable speed condition (Fig 7), and one outlier (10%) was observed in two subtasks of TUG under the fast speed condition (Fig 8).

## Discussion

The purpose of this study was to determine the concurrent validity and test-retest reliability of a video-based system developed to assess the movement speeds of nine TUG subtasks in

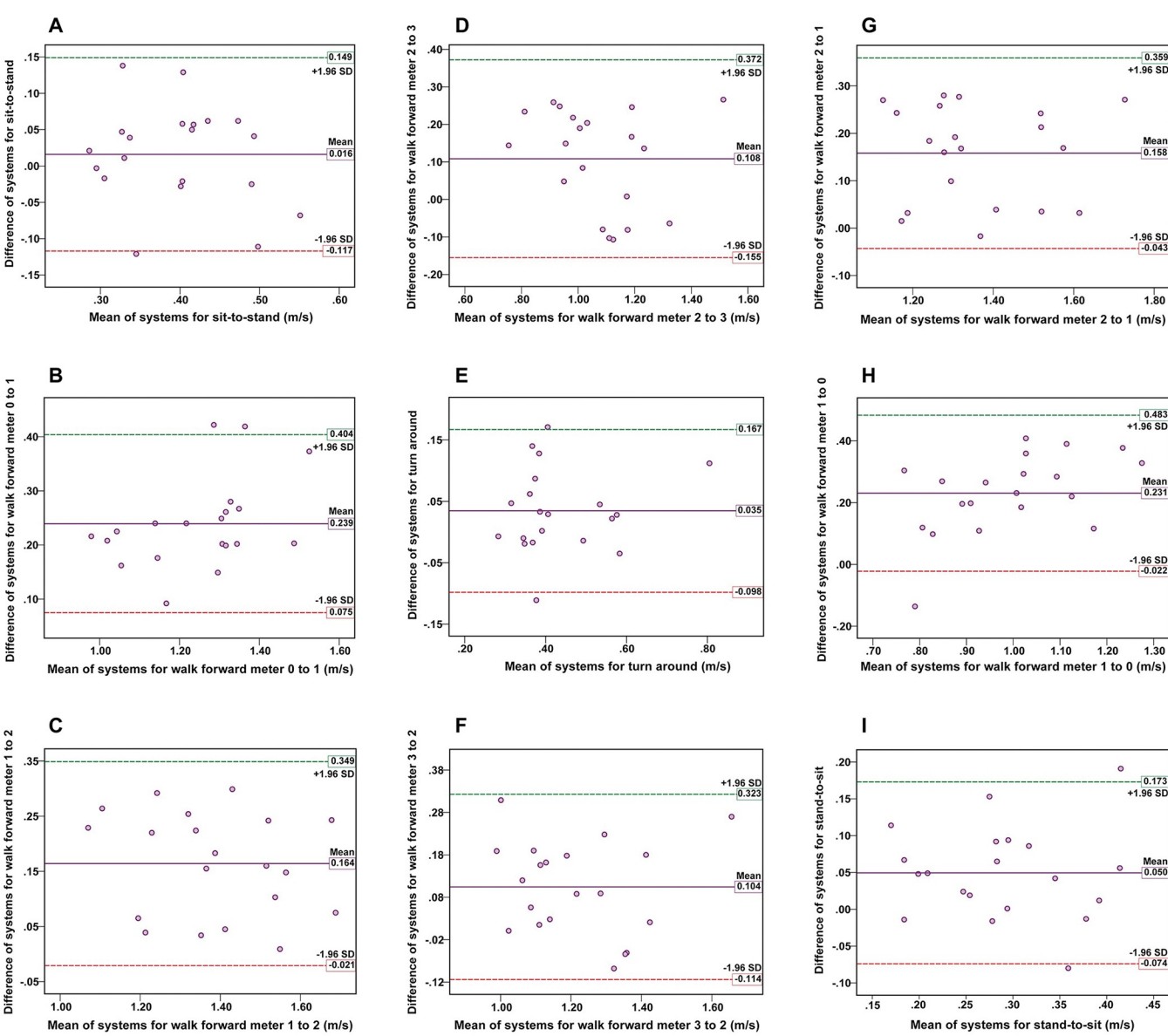

**Fig 6. The Bland-Altman plots in the concurrent validity assessment for the fast speed condition.** The plots display the agreement between the two systems for measurements of the subtask speed of the TUG: (A) sit-to-stand, (B) walk meter 0–1, (C) walk meter 1–2, (D) walk meter 2–3, (E) turn around, (F) walk meter 3–2, (G) walk meter 2–1, (H) walk meter 1–0, and (I) stand-to-sit. The x-axis represents the mean values, and the y-axis represents the mean difference between the two systems for each subtask of TUG. Reference lines show the mean difference between the two systems (solid line) and 95% LOA for the mean difference (dash line).

healthy older adults. We hypothesized that the video-based system developed would have high to very high concurrent validity and good to excellent test-retest reliability in all subtask movement speeds during both the comfortable and fast speed conditions. In line with our expectations, the movement speeds of all TUG subtasks showed high to very high degrees of correlation between the two systems excluding the sit-to-stand component, which showed a moderate correlation. In addition, the reliability of the video-based system was good to excellent in measuring each TUG subtask movement speed. To the best of our knowledge, this is the first study verifying the test-retest reliability of the conventional video-based system and its

**Table 3. Test-retest reliability of the video-based system for assessing the TUG subtasks.**

| TUG subtasks | Movement speed (m/s) | | ICC 3,2 (95% CI) | p-value |
|---|---|---|---|---|
| | Session 1 Mean (SD) | Session 2 Mean (SD) | | |
| **Comfortable speed** | | | | |
| $V_1$ (sit-to-stand) | 0.252 (0.56) | 0.261 (0.055) | 0.860 (0.445–0.965) | 0.004[a] |
| $V_2$ (walk meter 0–1) | 1.117 (0.161) | 1.162 (0.146) | 0.918 (0.659–0.980) | 0.0001[b] |
| $V_3$ (walk meter 1–2) | 1.182 (0.204) | 1.226 (0.192) | 0.957 (0.806–0.990) | 0.0001[b] |
| $V_4$ (walk meter 2–3) | 1.030 (0.168) | 1.041 (0.160) | 0.967 (0.872–0.992) | 0.0001[b] |
| $V_5$ (turn around) | 0.465 (0.066) | 0.454 (0.074) | 0.846 (0.383–0.962) | 0.006 [a] |
| $V_6$ (walk meter 3–2) | 1.052 (0.141) | 1.083 (0.161) | 0.957 (0.823–0.990) | 0.0001[b] |
| $V_7$ (walk meter 2–1) | 1.147 (0.168) | 1.160 (0.181) | 0.952 (0.812–0.988) | 0.0001[b] |
| $V_8$ (walk meter 1–0) | 0.888 (0.110) | 0.925 (0.142) | 0.851 (0.447–0.962) | 0.004[a] |
| $V_9$ (stand-to-sit) | 0.346 (0.093) | 0.287 (0.092) | 0.880 (0.498–0.971) | 0.003[a] |
| **Fast speed** | | | | |
| $V_1$ (sit-to-stand) | 0.294 (0.056) | 0.290 (0.052) | 0.817 (0.225–0.955) | 0.012[a] |
| $V_2$ (walk meter 0–1) | 1.396 (0.109) | 1.412 (0.093) | 0.908 (0.649–0.977) | 0.001[a] |
| $V_3$ (walk meter 1–2) | 1.490 (0.179) | 1.506 (0.160) | 0.979 (0.921–0.995) | 0.0001[b] |
| $V_4$ (walk meter 2–3) | 1.217 (0.122) | 1.223 (0.147) | 0.896 (0.570–0.974) | 0.002[a] |
| $V_5$ (turn around) | 0.474 (0.087) | 0.492 (0.069) | 0.822 (0.319–0.955) | 0.009[a] |
| $V_6$ (walk meter 3–2) | 1.279 (0.157) | 1.281 (0.146) | 0.948 (0.786–0.987) | 0.0001[b] |
| $V_7$ (walk meter 2–1) | 1.409 (0.175) | 1.419 (0.151) | 0.950 (0.802–0.988) | 0.0001[b] |
| $V_8$ (walk meter 1–0) | 1.124 (0.121) | 1.098 (0.079) | 0.720 (0.097–0.930) | 0.038[a] |
| $V_9$ (stand-to-sit) | 0.350 (0.111) | 0.350 (0.094) | 0.880 (0.498–0.971) | 0.003[a] |

ICC 3,2 = 2-way mixed effects intraclass correlation coefficient model.

[a]Significant at $p < 0.05$

[b]Significant at $p < 0.001$.

concurrent validity with the motion analysis system for detecting the movement speed of each subtask of TUG among older adults.

## Concurrent validity

The motion analysis system was used as a gold standard for measuring the TUG subtask movement speeds. In the present study, the movement speed of each subtask from the video-based system was obtained from 20 to 30 instantaneous speed values, while those from the motion analysis system were acquired from 70 to 100 instantaneous speed values. The findings from this study demonstrated that the correlations of our developed video-based system with the motion analysis system were high to very high in all subtasks across the two walking conditions, with the exception of the sit-to-stand subtask ($V_1$), which showed a moderate correlation at both comfortable and fast paces. Moreover, visual inspection of the Bland-Altman plots for the concurrent validity showed good agreement between the two systems, indicating that the two systems are partly interchangeable in measuring the movement speed of TUG subtasks.

Among the nine subtask movement speed correlations, the lowest correlation was observed for the sit-to-stand subtask. This may be due to the mismatch of the centroid position detection between the two systems. Regarding the background subtraction algorithm, the video-based program calculated participants' centroid position from a two-dimensional geometry intersection (a convergence between the horizontal back-to-leg line and the vertical head-to-foot line). When participants were in a sitting position, the centroid position would be more

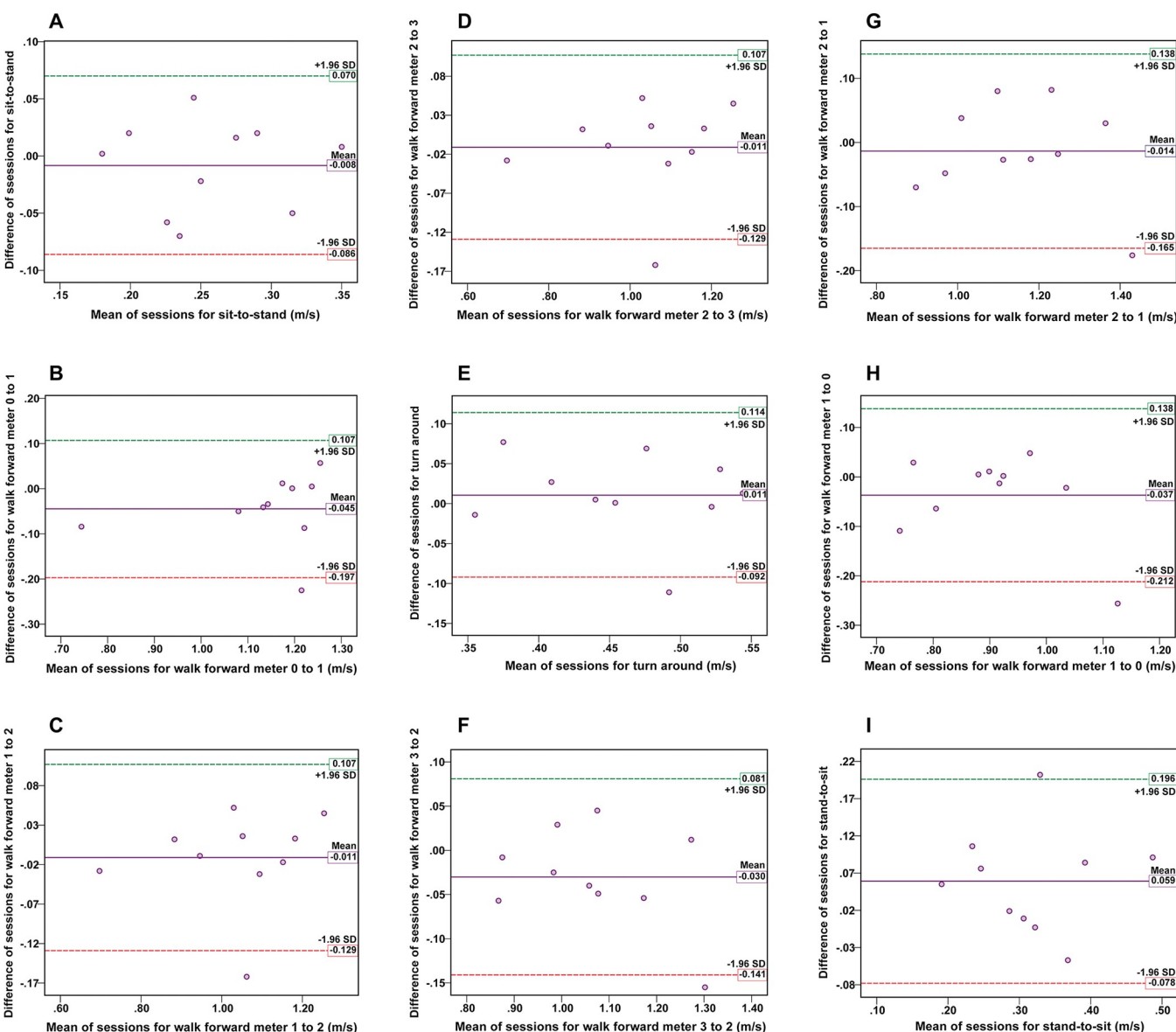

**Fig 7. The Bland-Altman plots in the test-retest reliability assessment for the comfortable speed condition.** The plots display the agreement between the two sessions for measurements of the subtask speed of the TUG: (A) sit-to-stand, (B) walk meter 0–1, (C) walk meter 1–2, (D) walk meter 2–3, (E) turn around, (F) walk meter 3–2, (G) walk meter 2–1, (H) walk meter 1–0, and (I) stand-to-sit. The x-axis represents the mean values, and the y-axis represents the mean difference between the two sessions for each subtask of TUG. Reference lines show the mean difference between the two sessions (solid line) and 95% LOA for the mean difference (dash line).

excessive than when in a standing position due to the system accounting for the thigh length. Therefore, participants' centroid position was not in the center of the trunk. Unlike the video-based system, the motion analysis system calculated the centroid position from a crossing line of ASIS and PSIS markers; thus, the centroid position was constantly in the center of the trunk. Differences in the centroid position detection between the two systems may affect the concurrent validity of the sit-to-stand subtask.

The present findings revealed that the walking forward ($V_2$, $V_3$, $V_4$) subtasks during walking at both the comfortable and fast speeds had the highest relative agreement between the two systems compared to other subtasks. It might be possible that the centroid position of each

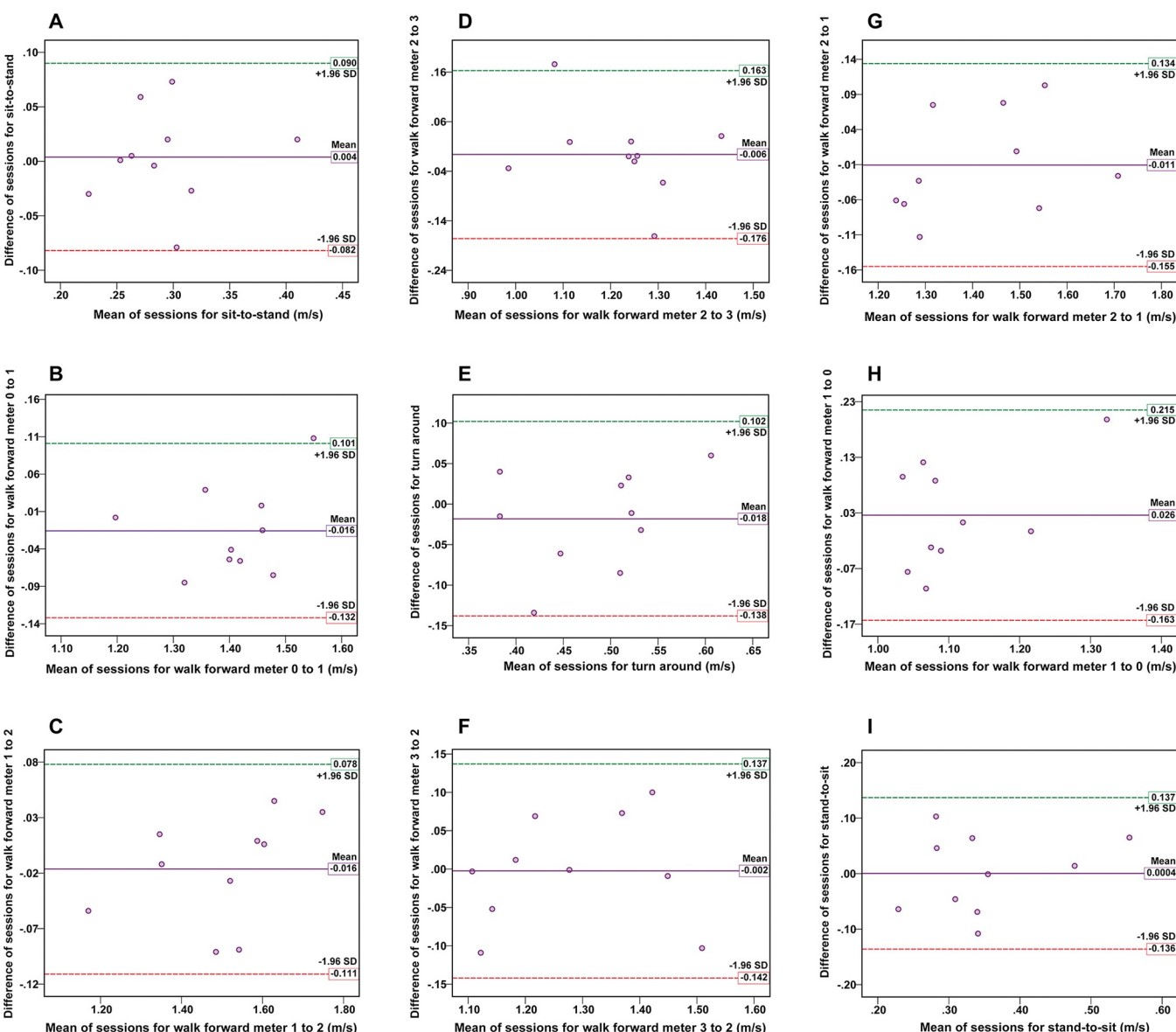

**Fig 8. The Bland-Altman plots in the test-retest reliability assessment for the fast speed condition.** The plots display the agreement between the two sessions for measurements of the subtask speed of the TUG: (A) sit-to-stand, (B) walk meter 0–1, (C) walk meter 1–2, (D) walk meter 2–3, (E) turn around, (F) walk meter 3–2, (G) walk meter 2–1, (H) walk meter 1–0, and (I) stand-to-sit. The x-axis represents the mean values, and the y-axis represents the mean difference between the two sessions for each subtask of TUG. Reference lines show the mean difference between the two sessions (solid line) and 95% LOA for the mean difference (dash line).

participant during the walking forward subtask moved in the line of horizontal plane calibration (*z*-axis) and covered the field of view of a two-dimensional camera. Accordingly, the forward walking speed is the most accurate and free from the effect of lens distortion. Our results are in agreement with previous studies that reported the high concurrent validity of the conventional video-based system against the motion analysis system in measuring the instantaneous gait speed during straight walking at a slow, usual, and fast pace in healthy older adults [25,26]. In addition, Aung et al. [27] supported that the simple video-based system had an excellent degree of correlation compared to the force distribution measurement platform in

investigating gait speed and spatiotemporal gait parameters in individuals with stroke when walking forward at preferred and fast speeds.

## Test-retest reliability

Although the number of participants in the reliability study was relatively small in the present study, the video-based system demonstrated good to excellent reliability for nine subtask movement speeds at both the comfortable and fast speeds. In addition, the Bland-Altman plots for test-retest reliability of the video-based system exhibited a narrow range of the LOA and the one outlier in some subtasks of TUG, suggesting a trivial natural variation over time. It might be possible that the average movement speed values in each TUG subtask were derived from several instantaneous speed values (20 to 30 values in each subtask), which enhanced a consistent reproduction of the result over two occasions. Another explanation is that the testing was administered under controlled laboratory conditions and scheduled participants individually; thus, participants performed the test with no distractions. Additionally, the older adults participating in the study were healthy and cognitively intact, and were allowed a practice trial; subsequently, they correctly followed the testing protocol. Moreover, the assessors were well-trained in the data collection procedures in terms of system set up, testing protocol, and standard instruction of TUG. Specifically, three assessors performed the same role throughout the data collection to enhance the intra-reliability of the study. Lastly, the video-based system is user-friendly because it is easy to set up and analyze the data. For data calculation, the movement speed in each TUG subtask was a semi-automated process using a MATLAB program. The time taken to complete the data calculation process for novice assessors was approximately 10 minutes. From all the standpoints raised, we describe the results of repeatability, which show good to excellent reliability over two measurement sessions. These basic findings are consistent with research showing that the video-based system was an excellent intratester reliability of gait speed and spatiotemporal parameters among individuals with stroke [26].

There are some limitations in the present study. The centroid position identification was inaccurate when participants were in the sitting position. As a result, future studies should develop a deep learning-based object detection algorithm to automatically capture participant's centroid position. Moreover, older adult participants in this study were healthy and had normal cognitive functioning, so these findings may not be generalizable to older people with health conditions. In addition, the small sample size may influence statistical power; thus, future studies with larger sample sizes would enhance the power analysis. Finally, our study administered the TUG test in a controlled laboratory rather than a free environment, which restricts the external validity and reliability of the video-based system. Assessment of the validity and reliability of the video-based system while performing the TUG test in free-living environments is warranted.

## Conclusions

The present study illustrated that the conventional video-based system had moderate to very high concurrent validity compared to the motion analysis system in measuring TUG subtask movement speeds when walking at both the comfortable and fast speeds. The test-retest reliability of the video-based system was good to excellent for all TUG subtasks under both the comfortable and fast speeds. The Bland-Altman plots showed that the video-based system had good agreement with the motion analysis system and high stability over time for measuring TUG subtask movement speeds among healthy older adults.

## Supporting information

**S1 File. TUG subtask movement speeds from the video-based system and motion analysis system (concurrent validity study).** Means of movement speed in nine TUG subtasks in both the comfortable and fast speed conditions from the video-based system and motion analysis system for each participant.
(PDF)

**S2 File. TUG subtask movement speeds from the video-based system in session 1 and session 2 (test-retest reliability).** Means of movement speed in nine TUG subtasks in both the comfortable and fast speed conditions from two sessions using the video-based system for each participant.
(PDF)

## Acknowledgments

The authors would like to thank Wanya N., Parajam F., Chotchaung W., and Phapatarinan K. for their contributions to the data collection. In addition, we are thankful to all staff of the sub-district administrative organization for their assistance with participant recruitment.

## Author Contributions

**Conceptualization:** Teerawat Kamnardsiri, Nuanlaor Thawinchai, Arisa Parameyong, Pim Pholjaroen, Khanittha Wonglangka, Paphawee Prupetkaew, Sirinun Boripuntakul.

**Data curation:** Teerawat Kamnardsiri, Nuanlaor Thawinchai, Arisa Parameyong, Pim Pholjaroen, Khanittha Wonglangka, Paphawee Prupetkaew, Sirinun Boripuntakul.

**Formal analysis:** Teerawat Kamnardsiri, Nuanlaor Thawinchai, Arisa Parameyong, Pim Pholjaroen, Khanittha Wonglangka, Paphawee Prupetkaew, Sirinun Boripuntakul.

**Funding acquisition:** Teerawat Kamnardsiri.

**Investigation:** Teerawat Kamnardsiri, Nuanlaor Thawinchai, Arisa Parameyong, Pim Pholjaroen, Khanittha Wonglangka, Paphawee Prupetkaew, Sirinun Boripuntakul.

**Methodology:** Teerawat Kamnardsiri, Nuanlaor Thawinchai, Arisa Parameyong, Pim Pholjaroen, Khanittha Wonglangka, Paphawee Prupetkaew, Sirinun Boripuntakul.

**Project administration:** Teerawat Kamnardsiri, Sirinun Boripuntakul.

**Resources:** Teerawat Kamnardsiri, Sirinun Boripuntakul.

**Software:** Teerawat Kamnardsiri.

**Supervision:** Sirinun Boripuntakul.

**Validation:** Teerawat Kamnardsiri, Sirinun Boripuntakul.

**Visualization:** Teerawat Kamnardsiri.

**Writing – original draft:** Teerawat Kamnardsiri, Sirinun Boripuntakul.

**Writing – review & editing:** Teerawat Kamnardsiri, Nuanlaor Thawinchai, Arisa Parameyong, Pim Pholjaroen, Khanittha Wonglangka, Paphawee Prupetkaew, Sirinun Boripuntakul.

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
