## [Decision Letter · Decision Letter 0]

30 Mar 2023

PONE-D-23-03799Conventional video-based system for measuring the subtask speed of the Timed Up and Go test in older adults: Validity and reliability studyPLOS ONE

Dear Dr. Boripuntakul,

Thank you for submitting your manuscript to PLOS ONE. After careful consideration, we feel that it has merit but does not fully meet PLOS ONE’s publication criteria as it currently stands. Therefore, we invite you to submit a revised version of the manuscript that addresses the points raised during the review process.

We look forward to receiving your revised manuscript.

Kind regards,

Ateeb Ahmad Parray, BDS, MSS, MPH

Academic Editor

PLOS ONE

Reviewers' comments:

Reviewer's Responses to Questions

**Comments to the Author**

1. Is the manuscript technically sound, and do the data support the conclusions?

Reviewer #1: Yes

Reviewer #2: Yes

2. Has the statistical analysis been performed appropriately and rigorously? 

Reviewer #1: Yes

Reviewer #2: Yes

3. Have the authors made all data underlying the findings in their manuscript fully available?

Reviewer #1: Yes

Reviewer #2: Yes

4. Is the manuscript presented in an intelligible fashion and written in standard English?

Reviewer #1: Yes

Reviewer #2: Yes

5. Review Comments to the Author

Reviewer #1: Congratulations to the authors of the manuscript. This is a validation study that aims to verify the reliability of the video-based system for measuring the timed up-and-go test. This is a usual assessment performed in older adults and it is important to further understand the alternative methods to evaluate in a simple manner. So, this is an interesting study to complement previous knowledge on this.

However, to perform validation, the study needs more participants. This study evaluated twenty older adults to validate the test and ten to verify the reliability. The small sample reduces the power of the analysis and this way, reduces the power of validation. This is a major issue of the current study. Some specific comments:

Abstract:

- The first part of the abstract (Background) can be reduced to be more concise.

- Introduction

Despite the relevance of the study explained, the authors should be more careful and specific in providing limitations of existing systems and advantages from the video-based system. For example, “simple to administer, easy to set up, and inexpensive” can be questionable.

Moreover, considering the evolution of methods and procedures, why should we need a validity assessment of conventional video-based systems?

Material and Methods

- The procedures are explained properly. Some information can be added though. In specific, the participants performed TUG with two different walking speeds, one of them, fast. Was fast speed monitored or feedback provided? Was it fast speed or maximal speed? Is the interval of 1min is enough to recover to perform another repetition? Why the authors used two repetitions at each speed? Three repetitions would allow more reliable data (for example, ICC values).

- Could the authors explain why a different number of participants were used to verify the test-retest reliability? And, why different intervals and sessions were used? (compared to the first test). Can this compromise result?

-

Results and discussion:

- Perhaps, Table 1 can be included in the methods, during the participants’ presentation. This is not mandatory. Moreover, I think that Table 1 can be more simple and shorter. For example, the gender ratio and fallen can be presented in written text and this column can be removed.

- The sit-to-stand test, in global, at both speeds, presented r=0.67/0.68, which is not a very high correlation between video systems and motion analysis.

- I suggest the authors include a Bland-Altman analysis (including Bland and Altman plot) to assess the agreement between the two methods.

- Moreover, relationships were determined but no comparisons were made. This should be added to help to validate methods. Considering that the sample is the same, the data cannot be different between both methods.

- Considering that only a part of the sample performed the re-test, I am not sure about the information that this analysis (test and re-test) adds to the manuscript. This should be clearly explained in the discussion.

Reviewer #2: The research paper is sound, well organized and well written. If focuses on an unobserved part of the famous TUG test which could subsequently lead to the cause of irregular TUG results. Some minor suggestions are:

- There should be a diagram summarizing a whole process of the video-based processing.

- In the equation 3-5, what is ni? Is it a typo which should be sum from i to n or not?

- The details of video processing environment e.g. computer and software specifications should be mentioned.

- Figure 2 should be also labeled with distances.

6. PLOS authors have the option to publish the peer review history of their article (what does this mean?). If published, this will include your full peer review and any attached files.

Reviewer #1: No

Reviewer #2: No

<quillbot-extension-portal></quillbot-extension-portal>

---

## [Author Response · Author response to Decision Letter 0]

14 Apr 2023

Response to Reviewers’ Comments

Journal Requirements:

Comment # 1. Please ensure that your manuscript meets PLOS ONE's style requirements, including those for file naming. The PLOS ONE style templates can be found at

Response # 1. We have rechecked and changed our manuscript according to PLOS ONE's style requirements.

Comment # 2. Please note that PLOS ONE has specific guidelines on code sharing for submissions in which author-generated code underpins the findings in the manuscript. In these cases, all author-generated code must be made available without restrictions upon publication of the work. Please review our guidelines at https://journals.plos.org/plosone/s/materials-and-software-sharing#loc-sharing-code and ensure that your code is shared in a way that follows best practice and facilitates reproducibility and reuse.

Response # 2. Thank you for your information.

Reviewers’ comments:

Reviewer's Responses to Questions

Reviewer # 1: 

Congratulations to the authors of the manuscript. This is a validation study that aims to verify the reliability of the video-based system for measuring the timed up-and-go test. This is a usual assessment performed in older adults and it is important to further understand the alternative methods to evaluate in a simple manner. So, this is an interesting study to complement previous knowledge on this.

Response: Thank you very much.

However, to perform validation, the study needs more participants. This study evaluated twenty older adults to validate the test and ten to verify the reliability. The small sample reduces the power of the analysis and this way, reduces the power of validation. This is a major issue of the current study. Some specific comments:

Abstract:

Comment # 1. The first part of the abstract (Background) can be reduced to be more concise.

Response # 1. Thank you for the suggestions. We have revised the background in the abstract section according to the reviewer’s suggestion (page 2, line 24-26) as follows:

“The Timed Up and Go Test (TUG) is a simple fall risk screening test that covers basic functional movement; thus, quantifying the subtask movement ability may provide a clinical utility. The video-based system allows individual’s movement characteristics assessment.”

Introduction:

Comment # 2. Despite the relevance of the study explained, the authors should be more careful and specific in providing limitations of existing systems and advantages from the video-based system. For example, “simple to administer, easy to set up, and inexpensive” can be questionable.

Moreover, considering the evolution of methods and procedures, why should we need a validity assessment of conventional video-based systems?

Response # 2. We have revised according to the reviewer’s comment as follows (page 3-4, line 66-72):

“Several recent studies have explored TUG subcomponents using sensor technology, such as wearable inertial sensors [9, 11], smartphones [11, 12], and ambient sensors [11, 13]. However, there are some challenges such as high costs, susceptibility to interference and damage, and limited range of use [9, 11-14]. The video-based system has received increasing attention in the movement analysis field. The video-based system, a markerless approach, is one of the more flexible ways of data acquisition that allows participants to move naturally under various environmental conditions.”

In the present study, the reason for validating the conventional video-based systems is that we aim to develop a new validation tool (by comparing it with the gold standard) for capturing each TUG subtask’s movement speeds in older adults. Previous studies have reported the high validity of the conventional video-based system against the motion analysis system for gait speed detection straightforward walking tasks (1-2). However, the validity of the conventional video-based system for measuring the subtask speed of the TUG is still warranted.

References:

1. Kamnardsiri T, Khuwuthyakorn P, Boripuntakul S. The development of a gait speed detection system for older adults using video-based processing. Proceedings of the 2019 4th International Conference on Biomedical Imaging, Signal Processing; 2019 Oct 17-19; Nagoya, Japan. New York, United States: Association for Computing Machinery; 2020. doi: 10.1145/3366174.3366190

2. Boripuntakul S, Panjaroen K, Kormkaew K, Yawisit P, Kamnardsiri T. Validity of a speed detection system for measuring gait speed in community-dwelling older adults. Proceedings of the 2019 3rd International Conference on Computational Biology and Bioinformatics; 2019 Oct 17; Nagoya, Japan. New York, United States: Association for Computing Machinery; 2019. doi: 10.1145/3365966.3365976

Material and Methods:

Comment # 3. The procedures are explained properly. Some information can be added though. In specific, the participants performed TUG with two different walking speeds, one of them, fast. 

Comment # 3.1. Was fast speed monitored or feedback provided? 

Response # 3.1. No, it was not. We administered the TUG according to standard protocol by instructing the participants to walk quickly as possible but safely.

Comment # 3.2. Was it fast speed or maximal speed? Is the interval of 1 min enough to recover to perform another repetition? 

Response # 3.2. It was a self-selected fast speed. Specifically, participants were self-determined to walk at the fast speed commonly used in daily activity. In addition, all participants were active and healthy older adults; thus, no fatigue was reported after testing in each trial. For this reason, the 1-minute rest between each trial was enough. 

Comment # 3.3. Why the authors used two repetitions at each speed? Three repetitions would allow more reliable data (for example, ICC values).

Response # 3.3. We used two repetitions in each speed for notifying the validity and reliability because we follow the previous studies protocol that reported a high validity of the video-based system in assessing gait speed in older adults (1-2). Moreover, the data set obtained from the video-based and motion analysis systems over two repetitions per condition was quite large, which allowed a lot of repetitive data comparison between the two systems.

References:

1. Kamnardsiri T, Khuwuthyakorn P, Boripuntakul S. The development of a gait speed detection system for older adults using video-based processing. Proceedings of the 2019 4th International Conference on Biomedical Imaging, Signal Processing; 2019 Oct 17-19; Nagoya, Japan. New York, United States: Association for Computing Machinery; 2020. doi: 10.1145/3366174.3366190

2. Boripuntakul S, Panjaroen K, Kormkaew K, Yawisit P, Kamnardsiri T. Validity of a speed detection system for measuring gait speed in community-dwelling older adults. Proceedings of the 2019 3rd International Conference on Computational Biology and Bioinformatics; 2019 Oct 17; Nagoya, Japan. New York, United States: Association for Computing Machinery; 2019. doi: 10.1145/3365966.3365976

Comment # 4. Could the authors explain why a different number of participants were used to verify the test-retest reliability? And, why different intervals and sessions were used? (compared to the first test). Can this compromise result?

Response # 4. The fewer participants for the test-retest reliability than the validity study due to only some participants being available and willing to participate. However, there were many data set comparisons over two sessions due to the nature of video-based system data being extracted as instantaneous speed values (20 to 30 values in each subtask); thus, there was enough data for repeatability analysis. As confirmed by the results, the good to excellent reliability of the video-based system was observed for all subtask movement speeds at both the comfortable and fast speeds.

Regarding the reliability study, there were two sessions (30-minute interval between sessions) of the TUG test for measuring the consistency of the video-based system over two separate occasions. Each session consisted of two walking conditions: 1) walking at a comfortable speed for two trials and 2) walking at a fast speed for two trials, with a 1-minute break between trials. In the present study, the protocol throughout each session and each walking trial were similar, which enhanced the stability of the test over time.

Results and discussion:

Comment # 5. Perhaps, Table 1 can be included in the methods, during the participants’ presentation. This is not mandatory. Moreover, I think that Table 1 can be more simple and shorter. For example, the gender ratio and fallen can be presented in written text and this column can be removed.

Response # 5. Thank you for the suggestions. We have revised the demographic results and Table 1 regarding the reviewer’s suggestion (page 15, line 318-323) as follows:

“Twenty participants (5 male and 15 female) with a mean age of 66.90 (5.51) years participated in the validity study. Twelve older adults reported no history of falls, while three participants reported falling once during the past year. Among twenty participants, ten participants (2 male and 8 female) with a mean age of 63.60 (2.95) years underwent the reliability study. Nine participants did not have a history of falls, whereas one had a single fall in the past year.”

Comment # 6. The sit-to-stand test, in global, at both speeds, presented r=0.67/0.68, which is not a very high correlation between video systems and motion analysis.

Response # 6. As the reviewer’s concern, among the nine TUG subtasks, the lowest correlation between the two systems in both comfortable (r = 0.672, p < 0.001) and fast speed (r = 0.681, p < 0.001) was found in the sit-to-stand subtask. However, the correlation between the two systems was statistical significance meaningful, indicating a moderate relative agreement with p < 0.001 (1).

Reference:

1. Mukaka MM. Statistics corner: a guide to appropriate use of correlation coefficient in medical research. Malawi Med J. 2012;24(3):69-71.

Comment # 7. I suggest the authors include a Bland-Altman analysis (including Bland and Altman plot) to assess the agreement between the two methods.

Response # 7. Thank you for the suggestions. We have included and added information about Bland-Altman analysis throughout the manuscript. 

Comment # 8. Moreover, relationships were determined but no comparisons were made. This should be added to help to validate methods. Considering that the sample is the same, the data cannot be different between both methods.

Response # 8. Thank you for your point of view. Based on your recommendation, we have compared the data obtained from the video-based and motion-analysis systems. Paired t-test revealed a statistical difference in all subtask speeds of the TUG between the two systems, except for the sit-to-stand subtask.

Given the research design used in the present study, the correlation coefficient is a statistical approach for revealing the actual relationship between a new test (i.e., video-based system) and a gold standard test (i.e., motion analysis system). Correlation coefficient analysis is a robust statistical approach to provide the direction and magnitude of the relationship, in which the change in the magnitude of one variable is associated with a change in the magnitude of another variable (1). Although paired t-test is ideal for evaluating a difference between two data sets, it does not reveal significance when the average paired difference closes to zero. Thus, this test is unsuitable for characterization of the measurement relationship in the present situation (2). For this reason, we decided not to include the comparison findings between the two systems in the manuscript.

References:

1. Schober P, Boer C, Schwarte LA. Correlation coefficients: Appropriate use and interpretation. Anesth Analg. 2018;126(5):1763-8.

2. Linnet K. Limitations of the paired t-test for evaluation of method comparison data. Clin Chem. 1999;45(2):314-5.

Comment # 9. Considering that only a part of the sample performed the re-test, I am not sure about the information that this analysis (test and re-test) adds to the manuscript. This should be clearly explained in the discussion.

Response # 9. 

According to the reviewer's suggestion, we have included more information in the discussion section as follows (page 25, line 481-488):

“Although the number of participants in the reliability study was relatively small in the present study, the video-based system demonstrated good to excellent reliability for nine subtask movement speeds at both the comfortable and fast speeds. In addition, the Bland-Altman plots for test-retest reliability of the video-based system exhibited a narrow range of the LOA and the one outlier in some subtasks of TUG, suggesting a trivial natural variation over time. It might be possible that the average movement speed values in each TUG subtask were derived from several instantaneous speed values (20 to 30 values in each subtask), which enhanced a consistent reproduction of the result over two occasions.”

We acknowledged that it is a limitation of the study and stated in the manuscript as follows (page 26, line 507-508):

“In addition, the small sample size may influence statistical power; thus, future studies with larger sample sizes would enhance the power analysis.”

Reviewer # 2: 

The research paper is sound, well organized and well written. If focuses on an unobserved part of the famous TUG test which could subsequently lead to the cause of irregular TUG results. 

Response: Thank you very much.

Some minor suggestions are:

Comment # 1. There should be a diagram summarizing a whole process of the video-based processing.

Response # 1. Thank you for the suggestions. We have included the diagram summarizing a whole process of the video-based processing in the methods section (Figure 2). 

Comment # 2. In the equation 3-5, what is ni? Is it a typo which should be sum from i to n or not?

Response # 2. 

Yes, it is a typo error, and we apologize for that. We have rewritten the equation 3-5 (page 8, line 176-182).

Comment # 3. The details of video processing environment e.g. computer and software specifications should be mentioned.

Response # 3. We have included the details of video processing environment regarding the reviewer’s comment as follows (page 12, line 246-257):

“The geometric centroid position of each participant and the TUG subtask segmentations were quantified using a MATLAB 2015a (The MathWorks, Inc., Natick, Massachusetts, USA) script with the computer vision and image processing toolbox. The Windows 10 OS laptop computer equipped with Intel (R) Core (TM) i5-8265U CPU@1.60 GHz, 2 GB NVIDIA graphic card, and 8 GB DDR4 RAM (ASUSTek Computer Inc., Taipei, Taiwan) was employed to calculate the data.”

Comment # 4. Figure 2 should be also labeled with distances.

Response # 4. We have revised Figure 2 regarding the reviewer’s suggestion. Please note we changed the caption from Figure 2 to Figure 3.

---

## [Decision Letter · Decision Letter 1]

19 May 2023

Conventional video-based system for measuring the subtask speed of the Timed Up and Go test in older adults: Validity and reliability study

PONE-D-23-03799R1

Dear Dr. Boripuntakul,

We’re pleased to inform you that your manuscript has been judged scientifically suitable for publication and will be formally accepted for publication once it meets all outstanding technical requirements.

Kind regards,

Eric R. Anson

Academic Editor

PLOS ONE

Additional Editor Comments (optional):

Reviewers' comments:

Reviewer's Responses to Questions

**Comments to the Author**

1. If the authors have adequately addressed your comments raised in a previous round of review and you feel that this manuscript is now acceptable for publication, you may indicate that here to bypass the “Comments to the Author” section, enter your conflict of interest statement in the “Confidential to Editor” section, and submit your "Accept" recommendation.

Reviewer #1: All comments have been addressed

2. Is the manuscript technically sound, and do the data support the conclusions?

Reviewer #1: Yes

3. Has the statistical analysis been performed appropriately and rigorously? 

Reviewer #1: Yes

4. Have the authors made all data underlying the findings in their manuscript fully available?

Reviewer #1: Yes

5. Is the manuscript presented in an intelligible fashion and written in standard English?

Reviewer #1: Yes

6. Review Comments to the Author

Reviewer #1: The authors answered properly and clarified each one of my comments. The manuscript was changed, accordingly.

7. PLOS authors have the option to publish the peer review history of their article (what does this mean?). If published, this will include your full peer review and any attached files.

Reviewer #1: No

---

## [Editor Report · Acceptance letter]

24 May 2023

PONE-D-23-03799R1 

Conventional video-based system for measuring the subtask speed of the Timed Up and Go Test in older adults: Validity and reliability study 

Dear Dr. Boripuntakul:

I'm pleased to inform you that your manuscript has been deemed suitable for publication in PLOS ONE. Congratulations! Your manuscript is now with our production department. 

Kind regards, 

on behalf of

Dr. Eric R. Anson 

Academic Editor

PLOS ONE